# Cryo-EM structure of VASH1-SVBP bound to microtubules

**Faxiang Li[1†], Yang Li[2†], Xuecheng Ye[2,3], Haishan Gao[1], Zhubing Shi[1], Xuelian Luo[1,2], Luke M Rice[2,3]\*, Hongtao Yu[1,4,5]\***

[1]Department of Pharmacology, University of Texas Southwestern Medical Center, Dallas, United States; [2]Department of Biophysics, University of Texas Southwestern Medical Center, Dallas, United States; [3]Department of Biochemistry, University of Texas Southwestern Medical Center, Dallas, United States; [4]Zhejiang Provincial Laboratory of Life Sciences and Biomedicine, School of Life Sciences, Westlake University, Hangzhou, China; [5]Institute of Biology, Westlake Institute for Advanced Study, Hangzhou, China

**\*For correspondence:**
Luke.Rice@UTSouthwestern.edu (LMR);
yuhongtao@westlake.edu.cn (HY)

[†]These authors contributed equally to this work

**Competing interests:** The authors declare that no competing interests exist.

**Abstract** The dynamic tyrosination-detyrosination cycle of α-tubulin regulates microtubule functions. Perturbation of this cycle impairs mitosis, neural physiology, and cardiomyocyte contraction. The carboxypeptidases vasohibins 1 and 2 (VASH1 and VASH2), in complex with the small vasohibin-binding protein (SVBP), mediate α-tubulin detyrosination. These enzymes detyrosinate microtubules more efficiently than soluble αβ-tubulin heterodimers. The structural basis for this substrate preference is not understood. Using cryo-electron microscopy (cryo-EM), we have determined the structure of human VASH1-SVBP bound to microtubules. The acidic C-terminal tail of α-tubulin binds to a positively charged groove near the active site of VASH1. VASH1 forms multiple additional contacts with the globular domain of α-tubulin, including contacts with a second α-tubulin in an adjacent protofilament. Simultaneous engagement of two protofilaments by VASH1 can only occur within the microtubule lattice, but not with free αβ heterodimers. These lattice-specific interactions enable preferential detyrosination of microtubules by VASH1.

## Introduction

Microtubules are dynamic cytoskeletal polymers that play pivotal roles in a wide variety of cellular processes in eukaryotes, including maintaining cell shape and polarity, facilitating cargo transport, and guiding chromosome segregation (*Dogterom and Koenderink, 2019*; *Janke and Magiera, 2020*). Microtubules are built from αβ-tubulin heterodimers that stack head-to-tail to form protofilaments, which interact laterally to form hollow tubules (*Brouhard and Rice, 2018*). The human genome encodes multiple α- and β-tubulin isotypes (*Gadadhar et al., 2017*). Post-translational modifications (PTMs) on various αβ-tubulin heterodimers, such as acetylation, palmitoylation, polyglycylation, polyglutamylation, and tyrosination-detyrosination, further diversify the functional properties of microtubules (*Janke and Magiera, 2020*; *Magiera et al., 2018*). The α- and β-tubulin isotypes combined with myriad PTMs constitute a 'tubulin code', which tunes the dynamics and partner binding of microtubules for diverse cellular functions (*Janke and Magiera, 2020*; *Magiera et al., 2018*).

Detyrosination is one of the first identified PTMs of tubulin (*Hallak et al., 1977*). The newly translated proteins of all α-tubulin isoforms except TUBA4A contain a tyrosine or phenylalanine at the very C-terminus, which can be cleaved by the recently identified carboxypeptidases, vasohibins 1 and 2 in complex with SVBP (VASH1/2-SVBP) (*Aillaud et al., 2017*; *Nieuwenhuis et al., 2017*; *Nieuwenhuis and Brummelkamp, 2019*). The tubulin tyrosine ligase (TTL) can re-ligate a tyrosine to the detyrosinated α-tubulin (*Ersfeld et al., 1993*). Detyrosination of α-tubulin regulates the functions of microtubules by altering interactions with microtubule-associated proteins (MAPs) and motors

(*Badin-Larçon et al., 2004*; *Barisic et al., 2015*; *McKenney et al., 2016*; *Peris et al., 2006*; *Peris et al., 2009*; *Sirajuddin et al., 2014*). For example, proteins containing the cytoskeleton-associated protein glycine-rich (CAP-Gly) domain, including CLIP-170 and p150[Glued], bind more efficiently to tyrosinated microtubules (*Badin-Larçon et al., 2004*; *Peris et al., 2006*).

The detyrosination-tyrosination cycle of α-tubulin and its dysregulation have been linked to several cellular, physiological, and pathological processes. For example, detyrosination of α-tubulin inhibits microtubule disassembly by suppressing the activity of depolymerizing motors, including mitotic centromere associated kinesin (MCAK) and KIF2A (*Peris et al., 2009*; *Webster et al., 1987*). The plus-end-directed kinetochore motor CENP-E prefers to bind to detyrosinated spindle microtubules, and this preference helps to guide the congression of pole-proximal chromosomes toward the equator during mitosis (*Barisic et al., 2015*). Finally, proper levels of detyrosinated microtubules in cardiomyocytes provide the necessary mechanical resistance and stiffness for functional contractility (*Chen et al., 2018*; *Robison et al., 2016*). Abnormally high levels of detyrosinated microtubules impair the contractility of cardiomyocytes. Dysregulation of the detyrosination-tyrosination cycle can also contribute to cancer and neurodegenerative disorders (*Magiera et al., 2018*; *Mialhe et al., 2001*). Loss-of-function mutations of SVBP in humans have been associated with brain abnormalities, including microcephaly, ataxia, and intellectual disability (*Iqbal et al., 2019*; *Pagnamenta et al., 2019*).

Tubulin-modifying enzymes often exhibit substrate specificity for free αβ-tubulin heterodimers or polymerized microtubules (*Janke and Magiera, 2020*). TTL exclusively modifies free αβ-tubulin heterodimers, as its interaction surface on αβ-tubulin is partially blocked in polymerized microtubules (*Prota et al., 2013*; *Szyk et al., 2011*). In contrast, VASH1/2-SVBP preferably detyrosinate polymerized microtubules (*Li et al., 2019*; *Nieuwenhuis et al., 2017*). The crystal structures of VASH1/2-SVBP have revealed how these enzymes recognize the C-terminal tyrosine (*Adamopoulos et al., 2019*; *Li et al., 2019*; *Liao et al., 2019*; *Wang et al., 2019*; *Zhou et al., 2019*), but the structural basis for their substrate specificity towards microtubules is not understood. In particular, it remains to be determined whether VASH1/2-SVBP make additional contacts with the microtubule lattice and whether they can distinguish the α-tubulin conformations in free αβ-tubulin heterodimers and polymerized microtubules.

Using cryo-electron microscopy (cryo-EM), we have determined the structure of GMPCPP-stabilized 14-protofilament human microtubules decorated with catalytically inactive human VASH1-SVBP. The C-terminal tail of α-tubulin engages the catalytic site of VASH1, indicating that the structure represents a productive enzyme-substrate docking complex. Aside from binding the C-terminal tail, VASH1 makes multiple contacts with the globular domain of α-tubulin and a second α-tubulin in an adjacent protofilament. Disruption of these contacts selectively impairs VASH1-mediated detyrosination of microtubules, but not that of αβ-tubulin or the C-terminal tail of α-tubulin. These additional contacts that are specific to the microtubule lattice thus underlie the substrate preference of VASH1/2-SVBP towards polymerized microtubules.

## Results

### Cryo-EM structure of VASH1-SVBP bound to GMPCPP-stabilized microtubules

We transfected HeLa cells with plasmids encoding Myc-tagged VASH1 and SVBP, and treated these cells for short durations with nocodazole or Taxol, which inhibited or promoted microtubule polymerization, respectively. The detyrosination levels of α-tubulin were greatly decreased by nocodazole but enhanced by Taxol (*Figure 1A–C*). These data confirmed the substrate preference of VASH1-SVBP for polymerized microtubules.

To elucidate the structural basis of this preference, we sought to determine the structure of VASH1-SVBP bound to microtubules using single-particle cryo-EM. Based on in vitro microtubule pelleting assays, the catalytically inactive C169S mutant of the protease domain of VASH1 (residues 52–310) in complex with SVBP interacted with GMPCPP-stabilized human microtubules that were polymerized from recombinant human αβ-tubulin heterodimers (*Figure 1D*). Raw cryo-EM images showed efficient decoration of microtubules by VASH1-SVBP C169S (*Figure 1—figure supplement 1A*). During data processing, poorly decorated microtubule particles were removed by several

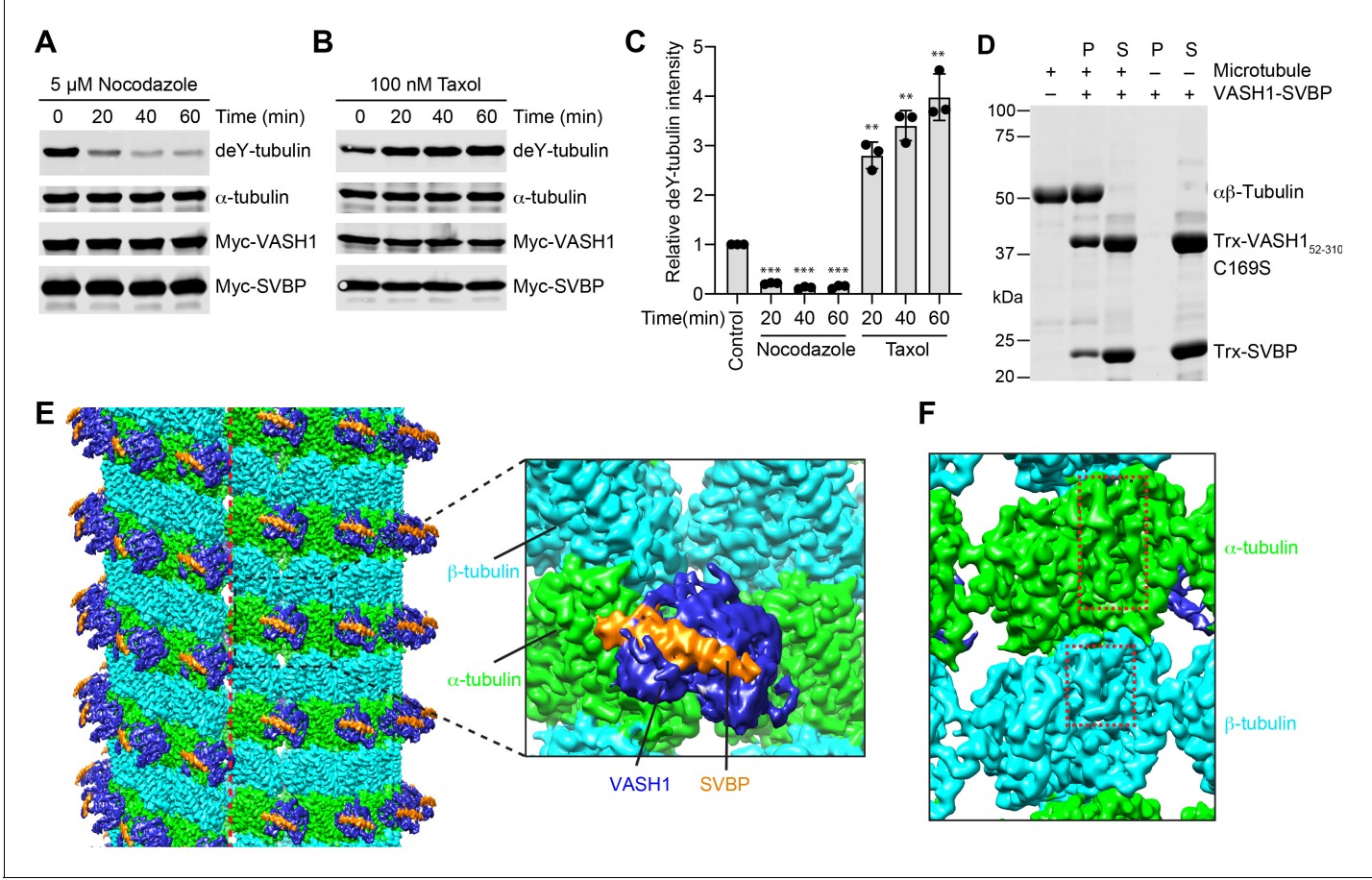

**Figure 1.** VASH1-SVBP efficiently detyrosinates and binds microtubules. (A,B) Tubulin detyrosination assays of VASH1-SVBP in human cells. HeLa Tet-On cells were co-transfected with VASH1 and SVBP plasmids, and treated with 5 μM nocodazole (A) or 100 nM Taxol (B) for indicated times at 24 hr post-transfection. The cell lysates were blotted with the indicated antibodies. deY-tubulin, detyrosinated α-tubulin. Experiments were repeated three times with similar results. (C) Quantification of the relative detyrosination levels of α-tubulin in (A) and (B) (mean ± s.d., n = 3 independent experiments). Significance calculated using two-tailed student's t-test; between control cells and cells treated with nocodazole or Taxol for the indicated time; *p < 0.05, **p < 0.01, ***p < 0.001, and ****p < 0.0001. (D) Microtubule pelleting assays showing the binding of VASH1-SVBP to recombinant human microtubules. S, supernatant; P, pellet. (E) Cryo-EM map of 14-protofilament, GMPCPP-stabilized microtubules decorated by the VASH1₅₂₋₃₁₀-SVBP complex. The catalytically inactive C169S mutant of VASH1 was used in the complex. The map is lowpass filtered to 4 Å. The microtubule seam is indicated by a red dashed line. α-tubulin, β-tubulin, VASH1, and SVBP are colored in green, cyan, blue, and orange, respectively. The same color scheme is used for all figures. The inset shows a close-up view of the boxed region. (F) Close-up view of the cryo-EM map in (E), viewed from the lumen. The α- and β-tubulin molecules can be distinguished by the length of the S9-S10 loop (boxed with red dashed lines), with the loop in α-tubulin being longer.

The online version of this article includes the following figure supplement(s) for figure 1:

**Figure supplement 1.** Structure determination of VASH1-SVBP-decorated GMPCPP-microtubules.

rounds of 2D classification (*Figure 1—figure supplement 1B*). Subsequent 3D classification revealed that, compared with 13-protofilament (PF) microtubules, 14-PF microtubules were the less populated but more ordered class. After seam search and 3D refinement, we determined the structure of 14-PF microtubules decorated with VASH1-SVBP to overall resolutions of 3.1 Å and 3.8 Å for microtubules and VASH1-SVBP, respectively (*Figure 1—figure supplement 1B*).

The EM map shows that VASH1-SVBP decorates the surface of 14-PF microtubules (*Figure 1E*). The seam of microtubules can be easily identified. At the seam, α-tubulin in one protofilament packs against β-tubulin in the protofilament across the seam. The α-tubulin and β-tubulin molecules can be assigned based on the length of the S9-S10 loop: this loop in α-tubulin is much longer than that in β-tubulin (*Figure 1F*). VASH1 simultaneously contacts two α-tubulin molecules in adjacent

protofilaments (*Figure 1E*). It has no contact with β-tubulin. SVBP does not appear to contribute directly to substrate recognition, because it has no contact with microtubules.

To build the structure of the microtubule-VASH1-SVBP complex, we docked the crystal structure of VASH1-SVBP and the cryo-EM structure of αβ-tubulin heterodimers of GMPCPP-stabilized microtubules into the EM density map as rigid bodies, and then manually adjusted the structures to better fit the density. The resolution of VASH1-SVBP is heterogenous, with the microtubule-binding regions and the core exhibiting higher resolution than the rest of the molecule (*Figure 2—figure supplement 1A*). With the current map, we could model all secondary structures of αβ-tubulin heterodimers except the C-terminal tails, and most of the side chains of αβ-tubulin and the bound nucleotides (*Figure 2A* and *Figure 2—figure supplement 1B,C*). By contrast, due to the relatively lower and heterogeneous resolution of VASH1-SVBP, only the side chains of residues that contact microtubules could be modeled (*Figure 2—figure supplement 1C*).

In the 6 Å lowpass-filtered EM map, the density of VASH1-SVBP is continuous, with most of the α-helices distinguishable (*Figure 2B*). The N- and C-terminal regions of SVBP remain disordered. Compared with crystal structures of VASH1-SVBP alone or bound to inhibitors, there are no obvious conformational changes of VASH1-SVBP upon binding to microtubules (*Figure 2—figure supplement 2A*). VASH1 simultaneously interacts with two α-tubulin molecules in adjacent protofilaments (*Figure 2C*). This binding mode can explain the substrate preference of VASH1 towards microtubules, because this collection of interactions is only possible within the microtubule lattice.

## Recognition of the α-tubulin C-terminal tail by VASH1

How vasohibins recognize the C-terminal tail of α-tubulin (CTα) is unresolved. Based on crystal structures of VASH1/2 bound to substrate-mimicking covalent inhibitors and subsequent mutagenesis data, we and two other groups identified the positively charged groove near the active site as the CTα-binding site (*Adamopoulos et al., 2019*; *Li et al., 2019*; *Wang et al., 2019*). A later study challenged this model, however, and showed that the free CTα peptide did not bind to the positively charged groove of VASH2, but instead bound to a site on the opposite side of the active site, with the backbone of CTα orientated in the opposite direction (*Zhou et al., 2019*).

Our 7 Å lowpass-filtered EM map shows an unaccounted, continuous density connecting the last modeled C-terminal residue of α-tubulin to the active site of VASH1 through the positively charged groove (*Figure 2D*). This weak density likely belongs to CTα, but the map quality of this region does not allow us to model the complete CTα. The recombinant human tubulin produced using the insect cell system contains human and insect α-tubulin molecules in roughly 70:30 proportion (*Ye et al., 2020*). Because the sequence of the C-terminal tail of the insect α-tubulin is different from that of human α-tubulin, it is possible that this heterogeneity contributes to the difficulty to resolve the α-tubulin tail.

The surface view of our microtubule-VASH1-SVBP structure reveals that the portion of the modeled CTα is located at one end of the groove and pointing towards the active site (*Figure 2—figure supplement 2B*). This binding mode is compatible for the detyrosination of α-tubulin. Thus, the structure captured in our study likely represents the catalysis-competent enzyme-substrate complex. We suspect that, because of its low sequence complexity (i.e. stretches of glutamates), the CTα peptide in isolation (i.e. not anchored to the globular domain of α-tubulin) may interact with VASH1 in purely electrostatic, nonproductive ways. Our results are consistent with the positively charged groove of VASH1/2 being the binding site of CTα, as we had originally proposed (*Li et al., 2019*).

## Interactions between VASH1 and microtubules

VASH1 contacts microtubules through three major interfaces (*Figure 3A*). Interfaces 1 and 2 are between VASH1 and the globular domain of the α-tubulin molecule that VASH1 would be acting on. Interface 3 is between VASH1 and a second α-tubulin molecule in an adjacent protofilament.

At interface 1, VASH1 residues on αD and the αC-αD loop form a hydrophilic interaction network with the C-terminal helix (α12) of α-tubulin (*Figure 3B*). Interface 2 is primarily formed between the β4-β5 loop of VASH1 and α11 of α-tubulin (*Figure 3C*). The β4-β5 loop of VASH1 is well-ordered, and has clear density in the EM map (*Figure 2—figure supplement 1C*). Residues in this VASH1 loop form hydrophobic and favorable electrostatic interactions with α-tubulin (*Figure 3C*).

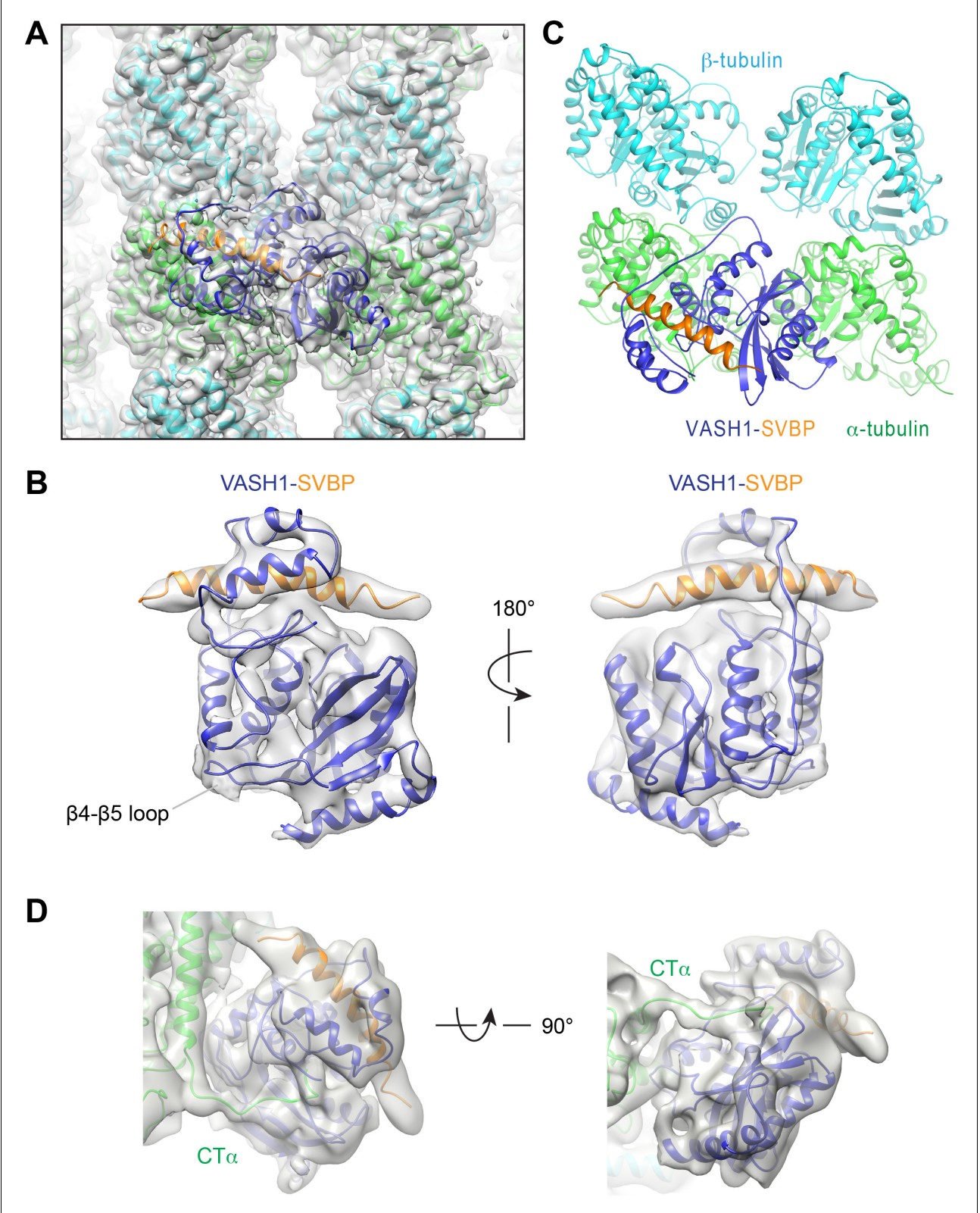

**Figure 2.** Cryo-EM structure of VASH1-SVBP bound to GMPCPP-stabilized microtubules. (A) Models of VASH1-SVBP (VASH1, blue; SVBP, orange) and tubulin (α-tubulin, green; β-tubulin, cyan) were docked into the cryo-EM density (lowpass-filtered to 4 Å) and refined. (B) Cryo-EM density map of VASH1-SVBP lowpass-filtered to 6 Å. (C) Ribbon diagram of the cryo-EM structure of VASH1-SVBP bound to microtubules. (D) Two views of the electron

*Figure 2 continued on next page*

*Figure 2 continued*

density map (generated by Phenix.auto_sharpen, with local B factor sharpening and resolution cutoff at 7 Å) showing an unfitted, continuous density that belonged to the α-tubulin C-terminal tail (CTα).

The online version of this article includes the following figure supplement(s) for figure 2:

**Figure supplement 1.** Cryo-EM map of VASH1-SVBP-decorated GMPCPP-microtubules.

**Figure supplement 2.** Interactions between microtubules and VASH1-SBVP.

Interface 3 lies between the last helix (αG) of VASH1 and a second α-tubulin molecule (termed α-tubulin') from the adjacent protofilament (*Figure 3D*). The total surface area buried between VASH1 and microtubules is 878 Å², 39.8% of which is contributed by interface 3. Because the density of VASH1 αG is poor (*Figure 2—figure supplement 1C*), we cannot definitively assign the precise orientation and conformation of the side chains in this helix. Nevertheless, R234, R296, R299, and L303 of VASH1 αG are located in close proximity to Y108, E155, R156, V159, E196, and E414 from α-tubulin'. These residues are likely to develop favorable electrostatic and hydrophobic interactions. Interface 3 can only be established between VASH1 and microtubules, but cannot exist between VASH1 and free αβ-tubulin heterodimers. Thus, the inter-protofilament interactions by VASH1 underlie the substrate preference of VASH1-SVBP towards microtubules.

## Contributions of VASH1-microtubule interactions to tubulin detyrosination

VASH1 residues that lie at the three VASH1-microtubule interfaces are conserved among VASH1 and VASH2 proteins from different species (*Figure 3—figure supplement 1A,B*), suggesting that they might be functionally important. To test the functional relevance of these interfaces, we introduced mutations into VASH1 to disrupt them, co-transfected these VASH1 mutants with SVBP into HeLa cells (which had low endogenous levels of tubulin detyrosination) (*Bulinski et al., 1988*), and evaluated their detyrosination activities on α-tubulin.

The single mutation of VASH1 R148 at interface 1 (R148E) greatly decreased α-tubulin detyrosination, and the K145E/R148E double mutation further attenuated the detyrosination activity of VASH1 (*Figure 4A* and *Figure 4—figure supplement 1A*). Interestingly, the VASH1 D155R mutant had higher detyrosination activity. D155 might develop unfavorable electrostatic interactions with E429 of α-tubulin (*Figure 3B*). Mutation of D155 to a positively charged residue, such as arginine, might have alleviated this clash, enhancing detyrosination. VASH1 mutations at interface 1 thus produce results consistent with structure-based predictions, validating its functional importance.

The VASH1 H268E/V270E and R234E/R299E/L303E (3E) mutations that targeted residues at interfaces 2 and 3, respectively, also reduced the detyrosination activity of VASH1 (*Figure 4A* and *Figure 4—figure supplement 1A*), consistent with the functional relevance of these interfaces as predicted from the structure. Because interface 3 only exists between VASH1 and microtubules, disrupting interface 3 via the VASH1 3E mutation should impair the detyrosination of microtubules, but not of free αβ tubulin. Consistent with this prediction, in cells treated with nocodazole, wherein microtubules are depolymerized, VASH1 WT and 3E had similar detyrosination activities, whereas mutations that targeted interfaces 1 and 2 still attenuated the detyrosination of free αβ-tubulin as predicted (*Figure 4B* and *Figure 4—figure supplement 1B*).

To further support the importance of the three contact sites, we expressed and purified recombinant VASH1 WT and mutants in complex with SVBP and tested their detyrosination activities on GMPCPP-stabilized human microtubules in vitro. Consistent with results in HeLa cells, mutations targeting interfaces 1–3 affected the detyrosination activity of VASH1 towards microtubules to varying degrees in vitro (*Figure 4C* and *Figure 4—figure supplement 1C*). Furthermore, these mutations had little effect on the detyrosination activity of VASH1 towards GST-CTα (*Figure 4D* and *Figure 4—figure supplement 1D*), ruling out the possibility that these mutations unintentionally affected the folding and structural integrity of VASH1. The VASH1 3E mutation that targets interface 3 had no effect on the detyrosination of free αβ-tubulin heterodimers in vitro (*Figure 4E* and *Figure 4—figure supplement 1E*). Finally, microtubule pelleting assays confirmed that the VASH1 mutants, K145E/R148E, H268E/V270E, and 3E, all had decreased binding to microtubules (*Figure 4F* and *Figure 4—*

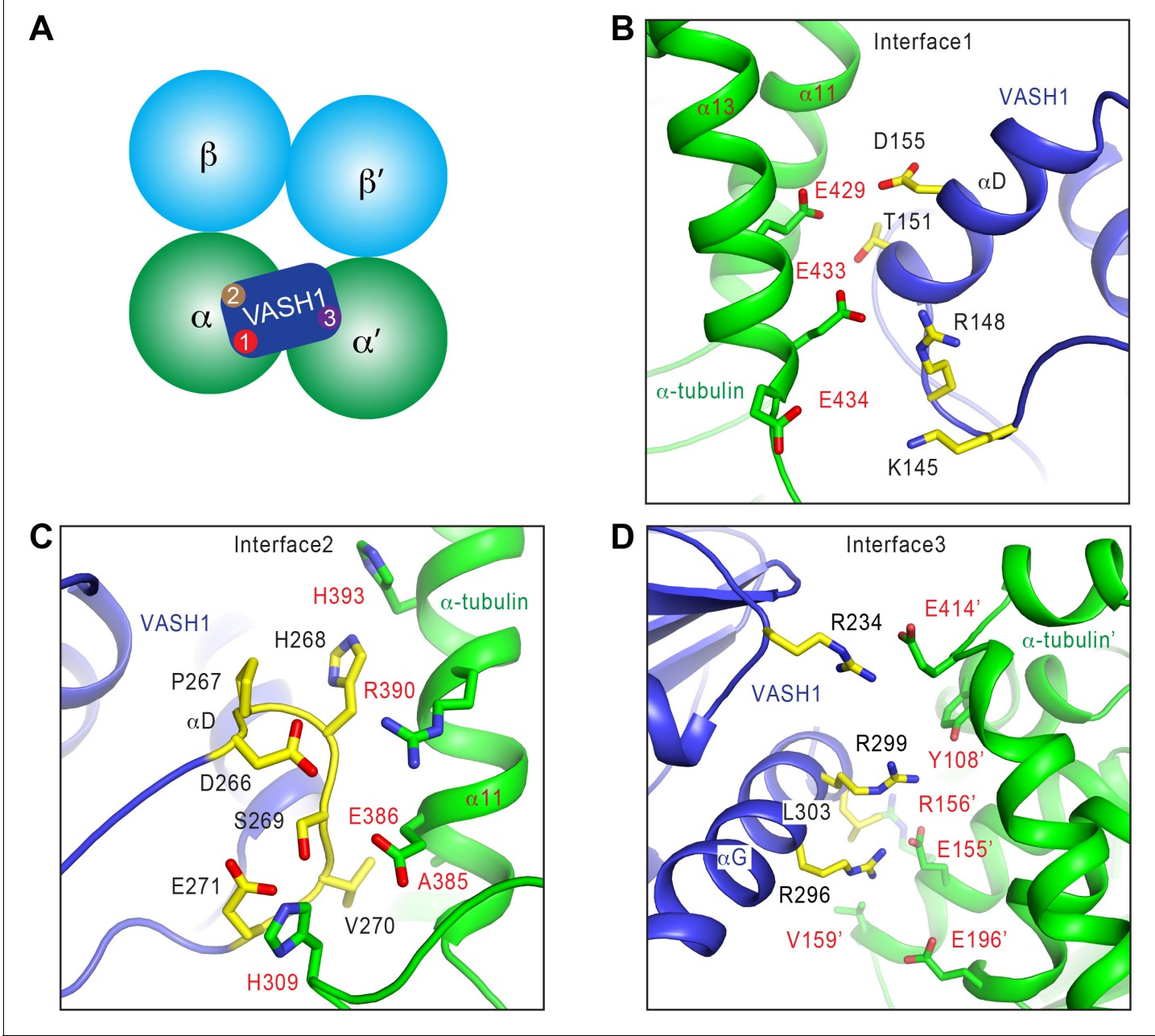

**Figure 3.** Interactions between VASH1 and microtubules. (**A**) Schematic drawing of the VASH1-microtubule complex, with the three main interfaces indicated. (**B–D**) Close-up views of the VASH1–microtubule interfaces 1 (**B**), 2 (**C**), and 3 (**D**), with interacting residues shown as sticks. VASH1 residues are colored yellow and labeled with black letters while α-tubulin residues are colored green and labeled with red letters.

The online version of this article includes the following figure supplement(s) for figure 3:

**Figure supplement 1.** Sequence alignment of VASH1 and VASH2 proteins.

*figure supplement 1F*). Collectively, these results establish the functional importance of the three VASH1-microtubule interfaces in microtubule binding and detyrosination.

## Discussion

VASH1/2-SVBP have recently been identified as the carboxypeptidases responsible for the cleavage of the last tyrosine of α-tubulin (*Aillaud et al., 2017*; *Nieuwenhuis et al., 2017*). Subsequent studies

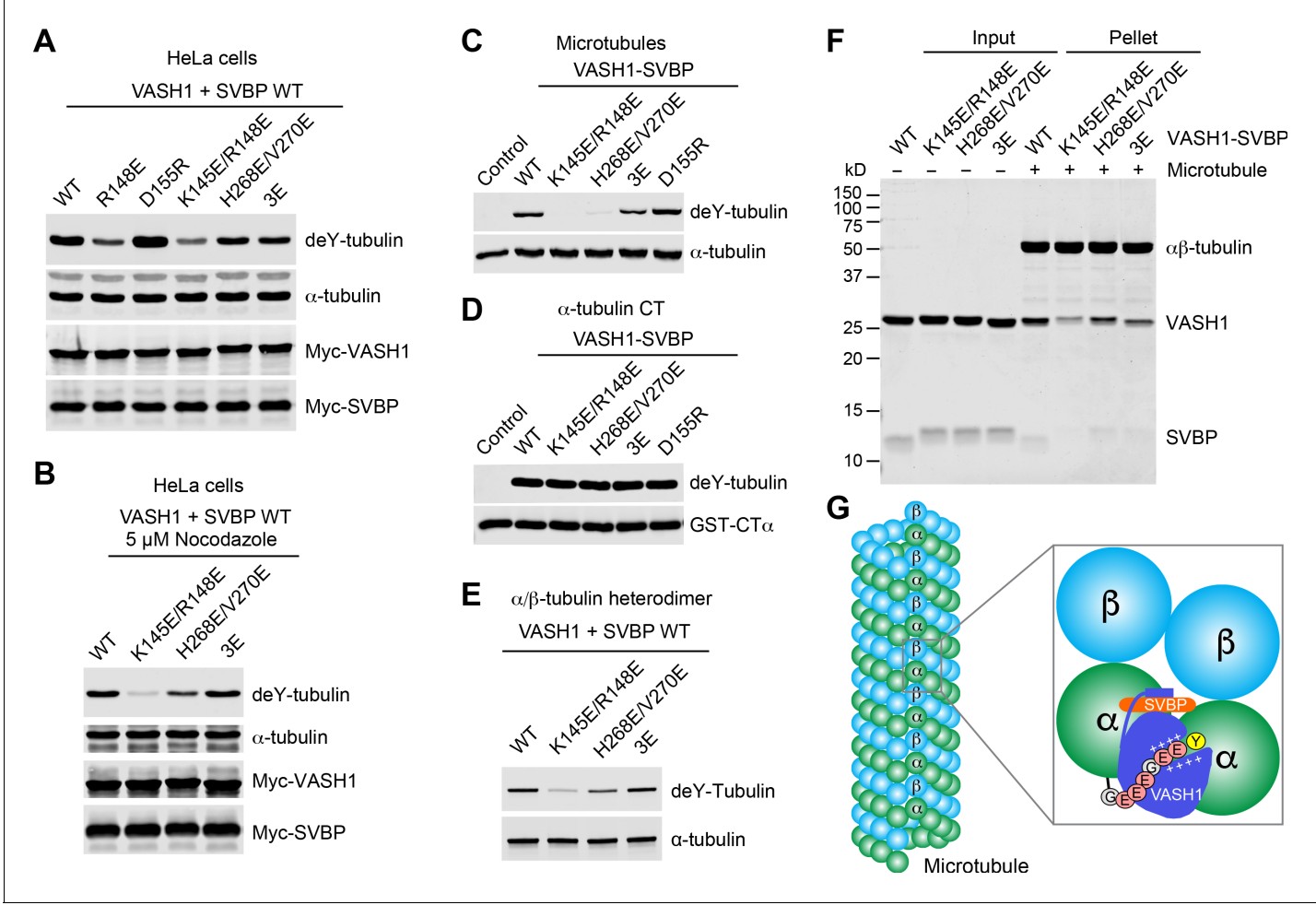

**Figure 4.** Requirement of VASH1-microtubule interactions in α-tubulin detyrosination. (**A,B**) Tubulin detyrosination assays of VASH1-SVBP WT or mutants in human cells. HeLa Tet-On cells were co-transfected with Myc-VASH1 WT or mutants and Myc-SVBP WT plasmids. At 24 hr post-transfection, the cells were treated without (**A**) or with 5 μM Nocodazole (**B**) for 1 hr. The cell lysates were blotted with the indicated antibodies. Compared with VASH1 WT, VASH1 mutants with multiple glutamate substitutions had slightly slower mobilities. The mobility shift is likely caused by the introduction of multiple negative charges, akin to protein phosphorylation, which also sometimes retards gel mobility. deY-tubulin, detyrosinated α-tubulin. 3E, R234E/R299E/L303E. Experiments were repeated three times with similar results. (**C–E**) In vitro detyrosination of GMPCPP-stabilized human microtubules (**C**), the C-terminal peptide of α-tubulin (CTα) fused to GST (**D**), or free αβ-tubulin heterodimers (**E**) by the indicated recombinant VASH1–SVBP WT or mutant complexes. Experiments were repeated at least three times with similar results. (**F**) Coomassie-stained gel of microtubule pelleting assays of VASH1-SVBP WT and mutant complexes. (**G**) Model of microtubule lattice binding, substrate recognition, and tubulin detyrosination by VASH1-SVBP. The '+' signs indicate positive charges.

The online version of this article includes the following figure supplement(s) for figure 4:

**Figure supplement 1.** Requirement of microtubule-VASH1 interactions in α-tubulin detyrosination.

have demonstrated the structural basis of the recognition of the C-terminal tyrosine by VASH1/2-SVBP (*Li et al., 2019*; *Liao et al., 2019*; *Wang et al., 2019*). VASH1/2-SVBP prefer polymerized microtubules as their substrate (*Li et al., 2019*; *Nieuwenhuis et al., 2017*), but the basis for this substrate preference has been unclear. Our study has now provided the structural basis for the substrate preference of VASH1/2-SVBP towards microtubules. VASH1-SVBP binds to the surface of polymerized microtubules and simultaneously engages two adjacent α-tubulin molecules in neighboring protofilaments through multiple interfaces (*Figure 4G*). We propose that these lattice-specific interactions guide the C-terminal tail of α-tubulin into the positively charged groove of VASH1 and position the C-terminal tyrosine into the active site for cleavage.

Because VASH1/2-SVBP prefer to detyrosinate microtubules and TTL prefers to act on αβ-tubulin heterodimers, the long-lived, stable microtubules in cells contain more detyrosinated α-tubulins,

whereas the newly formed, dynamic microtubules have more tyrosinated α-tubulin (*Gundersen et al., 1987*). Detyrosination of microtubules can alter the binding of MAPs, which indirectly affect microtubule dynamics. As microtubules assembled from tyrosinated and detyrosinated αβ-tubulins have similar intrinsic dynamic properties (*Janke and Magiera, 2020*), detyrosination per se does not directly stabilize microtubule dynamics, but rather it is a consequence of stable microtubules. On the other hand, VASH1-SVBP decorates the microtubule lattice very efficiently, and it straddles two neighboring protofilaments. It will be interesting to test whether, aside from detyrosination, VASH1/2-SVBP might directly bind to and stabilize microtubules.

In addition to interacting with and modifying microtubules, VASH1/2-SVBP are known to perform other functions. Indeed, VASH1/2-SVBP had been initially identified as secreted proteins that regulate angiogenesis (*Watanabe et al., 2004*). Recently, VASH1 has been shown to bind directly to the internal ribosome binding site (IRES) of mRNAs of angiogenesis factors and regulate their IRES-mediated translation in cardiomyocytes under hypoxic conditions (*Hantelys et al., 2019*). Biallelic loss-of-function mutations of *SVBP* in humans have been linked to brain development defects and intellectual disability (*Iqbal et al., 2019*; *Pagnamenta et al., 2019*). Although these patient phenotypes are likely caused by defective tubulin detyrosination and elevated levels of tubulin tyrosination in brain tissues (*Pagnamenta et al., 2019*), it is also possible that the angiogenesis and RNA-binding functions of VASH1/2-SVBP contribute to the disease phenotypes. In this study, using structure-based mutagenesis, we have identified VASH1 mutants that specifically disrupt microtubule binding and detyrosination. These mutants may prove valuable in future experiments aimed at dissecting the pathophysiological functions of vasohibins.

## Materials and methods

### Protein expression and purification
The expression and purification of the wild-type and mutant catalytic domain of human VASH1 (residues 52–310) bound to full-length SVBP and the C-terminal tail of human TUBA1A (residues 441–452) fused to GST (GST-CTα) were performed as previously described (*Li et al., 2019*). The recombinant human TUBA1B–TUBB3 heterodimer protein was expressed and purified as previously described (*Ti et al., 2016*; *Ye et al., 2020*).

### Preparation of cryo-EM grids
The TUBA1B–TUBB3 protein was thawed on ice and filtered through a 0.1 µm centrifugal Filter (EMD Millipore, UFC30VV00) at 4 ˚C to remove protein aggregates. The filtered protein was diluted to a final concentration of 15 µM in RBR110 buffer (110 mM PIPES pH 6.9, 1 mM MgCl$_2$, 1 mM EGTA) with 10 mM GMPCPP. The GMPCPP-stabilized microtubules were polymerized at 37 ˚C for 1 hr. The polymerized microtubules (4 µl) were applied to a glow discharged quantifoil holey carbon R1.2/1.3 copper grid (Electron Microscopy Sciences, Q350CR1.3) and allowed to adhere for 30 s in the chamber of a Vitrobot (ThermoFisher, Mark IV) set to 30 ˚C and 95% relative humidity. The VASH1-SVBP complex consisting of VASH1 C169S and full-length SVBP (2 µl of a 300 µM solution) was then added to the grid containing microtubules and incubated for 30 s. Then, 4 µl of the protein mixture was removed from the grid and another 2 µl of VASH1-SVBP was added. After another 30 s incubation, the grid was blotted for 4 s and plunged into ethane slush.

### Cryo-EM data collection
The dataset of GMPCPP-microtubules decorated with VASH1-SVBP C169S was collected with SerialEM on a 300-keV Titan Krios (FEI) transmission electron microscope. Movies were collected in super-resolution mode on a K3 Summit direct electron detector (Gatan), with a physical pixel size of 0.83 Å (nominal magnification 105,000X). The frame rate was 0.03 s/frame and the total exposure time was 1.2 s, resulting in an accumulated total exposure of ~50 electron/Å$^2$. A defocus range from −0.9 to −2.5 µm was used.

### Image processing
Motion correction for each movie stack was performed with the MotionCor2 program (*Zheng et al., 2017*). The contrast transfer function parameters were estimated from the motion-corrected

micrographs with Gctf (*Zhang, 2016*). The microtubules were selected manually from the motion-corrected micrographs in Relion (*Zivanov et al., 2018*). The selected microtubule images were computationally cut into overlapping boxes, with a ~80 Å non-overlapping region along the microtubule axis between adjacent boxes. 2D classification was performed in Relion to remove bad segments. Two rounds of helical refinement were performed in Relion, with each round of refinement assuming that microtubules all belonged 13- or 14-protofilament, respectively. The two resulting volumes were then used as templates in 3D classification to separate segments that belonged to 13- and 14-protofilament microtubules. Segments belonged to 14-protofilament microtubules were selected for an additional round of helical refinement in Relion to generate a map that did not distinguish α- and β-tubulins. The output map was low-pass filtered and masked to identify the seam and to impose the 1:1 VASH1:αβ-tubulin stoichiometry, and was used as the initial model for the following steps in Frealign (*Grigorieff, 2007*; *Grigorieff, 2016*). The segments were first aligned with global search, followed by local search with 14-fold pseudo-helical symmetry. After each iteration, a homemade script was used to crop out the best asymmetric unit and to rebuild the entire volume, which was used as the reference for the next iteration. The final resolution was estimated in Relion with the unfiltered half maps from the last iteration in Frealign.

## Model building, refinement and validation

The crystal structure of human VASH1-SVBP complex (PDB: 6OCG) and the EM structure of GMPCPP-stabilized microtubules (PDB: 6DPU) were fitted as rigid bodies into the cryo-EM map using the 'Fit in Map' utility in Chimera (*Pettersen et al., 2004*). A new αβ-tubulin model was generated in Coot (*Emsley and Cowtan, 2004*) by mutating residues of the GMPCPP-stabilized microtubules to match human TUBA1B and TUBB3 sequences and manually adjusted according to the density. For refinement, two copies of the αβ-tubulin model were docked into the segmented density from the final 14-protofilament map to form two adjacent tubulin heterodimers. Several rounds of further refinement were then performed using 'Real_space_refinement' in Phenix (*Adams et al., 2010*). The statistics and geometries for the final model were generated with MolProbity (*Williams et al., 2018*) and summarized in *Table 1*. The figures were prepared with PyMol (https://pymol.org/) or Chimera.

## Microtubule pelleting assays

The polymerized microtubules (with a monomer concentration of 15 μM) were prepared as described above for EM grid preparation. GMPCPP-stabilized microtubules and VASH1-SVBP wild type or mutants were mixed in pre-warmed RBR80 buffer with 5 mM GMPCPP, and incubated at 37 °C for 5 min. The final concentration of microtubules and VASH1-SVBP is 0.6 μM. The reaction mixtures were then centrifuged at 132,000 g for 30 min to pellet the microtubules. After three gentle washes with the warmed RBR80 buffer, the microtubule pellets were resuspended in the SDS sample buffer and analyzed by SDS-PAGE. The gels were stained by Coomassie brilliant blue and scanned with an Odyssey Infrared Imaging System (LI-COR).

## In vitro detyrosination assays

For in vitro detyrosination assays with GST-CTα (the C-terminal tail of α-tubulin) as the substrate, VASH1$_{52-310}$-SVBP wild type or mutants (100 nM) were incubated with 500 nM GST-CTα in the buffer containing 25 mM Tris, pH 7.5, 100 mM NaCl, and 1 mM DTT. The reaction mixtures were incubated at room temperature for 10 min. For in vitro detyrosination assays of microtubules, the polymerized αβ-tubulin proteins were diluted to 0.5 μM with warm BRB80 buffer. VASH1$_{52-310}$-SVBP wild type or mutant proteins (100 nM) were then incubated with 200 nM polymerized microtubules in the BRB80 buffer at room temperature for 5 min. The reactions were stopped by the addition of 2X SDS sample buffer, boiled, and subjected to SDS-PAGE followed by immunoblotting. The blots were scanned with an Odyssey Infrared Imaging System (LI-COR).

## Cell culture, transfection, and immunoblotting

HeLa Tet-On cells (Takara Bio USA, Inc) were used to analyze the detyrosination activities of VASH1-SVBP mutants because of the low endogenous detyrosination levels in these cells. The cell line has been validated to be of HeLa origin by short tandem repeat profiling. The cells are routinely checked

**Table 1.** Data collection and refinement statistics.

| | VASH1-SVBP-microtubule |
|---|---|
| Data collection and processing | |
| Magnification | 105,000 |
| Voltage (kV) | 300 |
| Electron exposure (e$^-$Å$^{-2}$) | 50 |
| Defocus range (μm) | −0.9 to −2.5 |
| Pixel size (Å) | 0.83 |
| Symmetry imposed | Pseudo-Helical |
| Initial particle images (no.) | 156,525 |
| Final particle images (no.) | 46,999 |
| Map resolution (Å)/FSC threshold | 3.1/0.143 |
| Map sharpening $B$ factor (Å$^{-2}$) | −60 |
| Refinement | |
| Initial model used | PDB: 6OCG, 6DPU |
| Model resolution (Å)/FSC threshold | 3.9/0.5 |
| Model composition | |
| Nonhydrogen atoms | 18,196 |
| Protein residues | 2296 |
| Ligands | 4 |
| $B$ factors (Å$^{-2}$) | |
| Protein | 143.68 |
| Ligands | 101.97 |
| R.m.s. deviations | |
| Bond lengths (Å) | 0.006 |
| Bond angles (°) | 0.931 |
| Validation | |
| MolProbity score | 1.81 |
| Clashscore | 6.2 |
| Poor rotamers (%) | 0.36 |
| Ramachandran plot | |
| Favored (%) | 92.56 |
| Allowed (%) | 7.35 |
| Disallowed (%) | 0.09 |

for mycoplasma contamination with DAPI staining. The cells were grown in DMEM (Invitrogen) supplemented with 10% fetal bovine serum and 2 mM L-glutamine. All plasmids for mammalian cell expression used in this study were cloned into the modified pCS2 vector that encodes six copies of the Myc tag at the N-terminus. Plasmid transfection was performed using the Lipofectamine 2000 Transfection Reagent (Thermo Fisher Scientific, 11668019) per the manufacturer's protocols. Cells in each well of a 12-well plate were transfected with a total of 1 μg plasmids (0.5 μg VASH1 and 0.5 μg SVBP) when the cell density reached 70% confluency. The cells were washed once with 1 ml PBS at 24 h post-transfection and collected by directly re-suspending them in 150 μl 1X SDS sample buffer. The samples were boiled at 100 °C for 15 min and subjected to immunoblotting with appropriate antibodies. The primary antibodies were used at a final concentration of 1 μg ml$^{-1}$ diluted in TBS containing 0.05% Tween 20% and 5% dry milk. Anti-mouse IgG (H+L) (Dylight 680 conjugates) and anti-rabbit IgG (H+L) (Dylight 800 conjugates) (Cell Signaling) were used as secondary antibodies. The blots were scanned with an Odyssey Infrared Imaging System (LI-COR).

## Acknowledgements

We thank D Nicastro and D Stoddard for cryo-EM facility access and data acquisition. Single particle cryo-EM data were collected at University of Texas Southwestern Medical Center (UTSW) Cryo-Electron Microscopy Facility, which is funded by a Cancer Prevention and Research Institute of Texas (CPRIT) Core Facility Support Award (Grant no. RP170644). This study is supported by grants from the National Institutes of Health (Grant nos. GM107415 to XL and GM098543 to LMR), Cancer Prevention and Research Institute of Texas (Grant nos. RP160255 to XL and RP160667-P2 to HY) and the Welch Foundation (Grant nos. I-1932 to XL, I-1908 to LMR, and I-1441 to HY).

## Additional information

### Funding

| Funder | Grant reference number | Author |
|---|---|---|
| National Institutes of Health | GM107415 | Xuelian Luo |
| National Institutes of Health | GM098543 | Luke M Rice |
| Cancer Prevention and Research Institute of Texas | RP160255 | Xuelian Luo |
| Cancer Prevention and Research Institute of Texas | RP160667-P2 | Hongtao Yu |
| Welch Foundation | I-1932 | Xuelian Luo |
| Welch Foundation | I-1908 | Luke M Rice |
| Welch Foundation | I-1441 | Hongtao Yu |

The funders had no role in study design, data collection and interpretation, or the decision to submit the work for publication.

### Author contributions

Faxiang Li, Yang Li, Conceptualization, Data curation, Formal analysis, Validation, Investigation, Visualization, Methodology, Writing - original draft; Xuecheng Ye, Resources, Investigation; Haishan Gao, Zhubing Shi, Data curation, Formal analysis; Xuelian Luo, Conceptualization, Writing - review and editing; Luke M Rice, Conceptualization, Data curation, Formal analysis, Supervision, Funding acquisition, Methodology, Writing - review and editing; Hongtao Yu, Conceptualization, Supervision, Funding acquisition, Project administration, Writing - review and editing

### Author ORCIDs

Faxiang Li (ID) https://orcid.org/0000-0002-6442-9063
Haishan Gao (ID) http://orcid.org/0000-0002-4954-8793
Zhubing Shi (ID) http://orcid.org/0000-0002-9624-4960
Xuelian Luo (ID) http://orcid.org/0000-0002-5058-4695
Luke M Rice (ID) https://orcid.org/0000-0001-6551-3307
Hongtao Yu (ID) https://orcid.org/0000-0002-8861-049X

### Decision letter and Author response

Decision letter https://doi.org/10.7554/eLife.58157.sa1
Author response https://doi.org/10.7554/eLife.58157.sa2

## Additional files

### Supplementary files

• Transparent reporting form

## Data availability

Coordinates and EM density maps have been deposited into the Protein Data Bank under the accession code 6WSL and EMD-21893, respectively.

The following datasets were generated:

| Author(s) | Year | Dataset title | Dataset URL | Database and Identifier |
|---|---|---|---|---|
| Li F, Yang L, Ye X, Gao H, Shi Z, Luo X, Rice LM, Yu H | 2020 | Cryo-EM structure of VASH1-SVBP bound to microtubules | http://www.rcsb.org/structure/6WSL | RCSB Protein Data Bank, 6WSL |
| Li F, Yang L, Ye X, Gao H, Shi Z, Luo X, Rice LM, Yu H | 2020 | Cryo-EM structure of VASH1-SVBP bound to microtubules | http://www.ebi.ac.uk/pdbe/entry/emdb/EMD-21893 | Electron Microscopy Data Bank, EMD-21893 |

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

# Appendix 1

**Appendix 1—key resources table**

| Reagent type (species) or resource | Designation | Source or reference | Identifiers | Additional information |
|---|---|---|---|---|
| Strain, strain background (*Escherichia coli*) | BL21(DE3) | Novagen | Cat.#: 69450 | competent cells |
| Cell line (*Homo-sapiens*) | HeLa Tet-On cells | Takara Bio | Cat.#: 631183 RRID:CVCL_V353 | Cell based detyrosination assay |
| Transfected construct (*Homo-sapiens*) | pCS2-MYC-human VASH1 WT | This paper | | Cell based detyrosination assay |
| Transfected construct (*Homo-sapiens*) | pCS2-MYC-human VASH1 R148E | This paper | | Cell based detyrosination assay |
| transfected construct (*Homo-sapiens*) | pCS2-MYC-human VASH1 D155R | This paper | | Cell based detyrosination assay |
| Transfected construct (*Homo-sapiens*) | pCS2-MYC-human VASH1 K145E/R148E | This paper | | Cell based detyrosination assay |
| Transfected construct (*Homo-sapiens*) | pCS2-MYC-human VASH1 H268E/V270E | This paper | | Cell based detyrosination assay |
| Transfected construct (*Homo-sapiens*) | pCS2-MYC-human VASH1 R234E/R299E/L303E | This paper | | Cell based detyrosination assay |
| Transfected construct (*Homo-sapiens*) | pCS2-MYC-human SVBP | This paper | | Cell based detyrosination assay |
| Antibody | anti-Myc (Mouse monoclonal) | Roche | Cat.#: 11667203001, RRID:AB_390911 | WB (1:5000) |
| Antibody | anti-$\alpha$-tubulin (Mouse monoclonal) | Sigma-Aldrich | Cat.#: T6199, RRID:AB_477583 | WB (1:2000) |
| Antibody | anti-detyrosinated tubulin (rabbit Polyclonal) | EMD Millipore | Cat.#: AB320, RRID:AB_177350 | WB (1:2000) |
| Antibody | anti-GST (Mouse monoclonal) | Sigma-Aldrich | Cat.#: SAB4200237, RRID:AB_2858197 | WB (1:2000) |
| Antibody | anti-rabbit IgG (H+L) (Dylight 800 conjugates) | Cell signaling | Cat.#:5151 RRID:AB_10697505 | WB (1:5000) |
| Antibody | anti-mouse IgG (H+L) (Dylight 680 conjugates) | Cell signaling | Cat.#: 5470 RRID:AB_10696895 | WB (1:5000) |
| Recombinant DNA reagent | pRSF-32M-3C-VASH1$_{52-310}$ WT | This paper | | See Materials and methods, Section: Protein expression and purification |

*Continued on next page*

*Appendix 1—key resources table continued*

| Reagent type (species) or resource | Designation | Source or reference | Identifiers | Additional information |
|---|---|---|---|---|
| Recombinant DNA reagent | pRSF-32M-3C-VASH1$_{52–310}$ C169S | This paper | | See Materials and methods, Section: Protein expression and purification |
| Recombinant DNA reagent | pRSF-32M-3C-VASH1$_{52–310}$ K145E/R148E | This paper | | See Materials and methods, Section: Protein expression and purification |
| Recombinant DNA reagent | pRSF-32M-3C-VASH1$_{52–310}$ H268E/V270E | This paper | | See Materials and methods, Section: Protein expression and purification |
| Recombinant DNA reagent | pRSF-32M-3C-VASH1$_{52–310}$ R234E/R299E/L303E | This paper | | See Materials and methods, Section: Protein expression and purification |
| Recombinant DNA reagent | pRSF-32M-3C-VASH1$_{52–310}$ D155R | This paper | | See Materials and methods, Section: Protein expression and purification |
| Recombinant DNA reagent | pET-32M-3C-SVBP | This paper | | See Materials and methods, Section: Protein expression and purification |
| Recombinant DNA reagent | pET-21b-SVBP | This paper | | See Materials and methods, Section: Protein expression and purification purification |
| Sequence-based reagent | VASH1_1up *Fse1* sense | This paper | PCR primers | GGAGGCCGGCCAATGCCAGGGGGGAAGAAG |
| Sequence-based reagent | VASH1_52 up *Fse1* sense | This paper | PCR primers | CGAGGCCGGCCAGACCTGCGAGACGGAGGC |
| Sequence-based reagent | VASH1_310 down *Asc1* anti-ense | This paper | PCR primers | CCAGGCGCGCCCTAGACCCGGATCTGGTACCC |
| Sequence-based reagent | VASH1_365 down *Asc1* anti-ense | This Paper | PCR primers | CCAGGCGCGCCCTAGACCCGGATCTGGTACCC |
| Sequence-based reagent | SVBP_1up *Fse1* sense | This Paper | PCR primers | TGCGGCCGGCCAATGGATCCACCTGCACGT |
| Sequence-based reagent | SVBP_66 down *Asc1* anti-sense | This Paper | PCR primers | CGTGGCGCGCCTCATTCTCCAGGAGGCTGC |
| Sequence-based reagent | VASH1_C169S anti-sense | This Paper | PCR primers | TTTGATTGGCAGGGCCTCT |
| Sequence-based reagent | VASH1_C169S sense | This Paper | PCR primers | AGCCTGGAAGCCGTGATCC |
| Sequence-based reagent | VASH1 K145E anti-sense | This Paper | PCR primers | AATTTCAAAGAACTGTGTCCCTGT |
| Sequence-based reagent | VASH1 K145E sense | This Paper | PCR primers | GAGAAGAGCAGACCTCTGACAGG |
| Sequence-based reagent | VASH1 R148E anti-sense | This Paper | PCR primers | GCTCTTCTTAATTTCAAAGAACTGT |
| Sequence-based reagent | VASH1 R148E sense also used for K145E/R148E mutation | This Paper | PCR primers | GAACCTCTGACAGGGCTGATG |

*Continued on next page*

*Appendix 1—key resources table continued*

| Reagent type (species) or resource | Designation | Source or reference | Identifiers | Additional information |
|---|---|---|---|---|
| Sequence-based reagent | VASH1 K145E/R148E anti-sense | This Paper | PCR primers | GCTCTTCTCAATTTCAAAGAACTGT |
| Sequence-based reagent | VASH1_D155R sense | This Paper | PCR primers | AGGGCTGATGCGCCTGGCCAAGG |
| Sequence-based reagent | VASH1_D155R anti-sense | This Paper | PCR primers | GTCAGAGGTCTGCTCTTC |
| Sequence-based reagent | VASH1_R234E sense | This Paper | PCR primers | GCCCGCCTTCGAGACGCTCAGCG |
| Sequence-based reagent | VSH1_R234E anti-sense | This Paper | PCR primers | GGCTTGTACATCAGGTCC |
| Sequence-based reagent | VASH1_268/270E sense | This Paper | PCR primers | CGAGGAGCAGATCGAGTGGAAGCAC |
| Sequence-based reagent | VASH1_268/270E anti-sense | This Paper | PCR primers | CTCTCCGGGTCGTGTGACACGCT |
| Sequence-based reagent | VASH1_R299E sense | This Paper | PCR primers | GCGCCACGCCGAGGACATGCGGC |
| Sequence-based reagent | VASH1_R299E anti-sense | This Paper | PCR primers | TCCAGCTCCTTGCGGAAG |
| Sequence-based reagent | VASH1_L303E sense | This Paper | PCR primers | CGACATGCGGGAGAAGATTGGCAAAGGGACGGGC |
| Sequence-based reagent | VASH1_L303E anti-sense | This Paper | PCR primers | CGGGCGTGGCGCTCCAGC |
| Peptide, recombinant protein | VASH1$_{52-310}$ WT in complex with SVBP | This paper | | In vitro detyrosination and pelleting assay |
| Peptide, recombinant protein | VASH1$_{52-310}$ D155R in complex with SVBP | This paper | | In vitro detyrosination |
| Peptide, recombinant protein | VASH1$_{52-310}$ K145E/R148E in complex with SVBP | This paper | | In vitro detyrosination and pelleting assay |
| Peptide, recombinant protein | VASH1$_{52-310}$ H268E/V270E in complex with SVBP | This paper | | In vitro detyrosination and pelleting assay |
| Peptide, recombinant protein | VASH1$_{52-310}$ R234E/R299E/L303E in complex with SVBP | This paper | | In vitro detyrosination and pelleting assay |
| chemical compound, drug | GMPCPP | Jena Bioscience | Cat.#: NC0641143 | |
| chemical compound, drug | Taxol | Cytoskeleton | Cat.#: TXD01 | |
| chemical compound, drug | Nocodazole | Sigma-Aldrich | Cat. #: M1404 | |
| chemical compound, drug | Isopropyl-beta-D-thiogalactoside (IPTG) | Gold Biotechnology | Cat. #: 12481C100 | Induce protein expression |
| Software, algorithm | UCSF Chimera | *Pettersen et al., 2004* | RRID:SCR_004097 | https://www.cgl.ucsf.edu/chimera/ |
| Software, algorithm | MotionCorr2 | *Zheng et al., 2017* | RRID:SCR_016499 | http://msg.ucsf.edu/em/software/motioncor2.html |
| Software, algorithm | GCTF | *Zhang, 2016* | RRID:SCR_016500 | https://www.mrc-lmb.cam.ac.uk/kzhang/Gctf/ |

*Continued on next page*

*Appendix 1—key resources table continued*

| Reagent type (species) or resource | Designation | Source or reference | Identifiers | Additional information |
|---|---|---|---|---|
| Software, algorithm | RELION3 | *Zivanov et al., 2018* | RRID:SCR_016274 | https://www3.mrc-lmb.cam.ac.uk/relion/index.php/Download_%26_install |
| Software, algorithm | Coot | *Emsley and Cowtan, 2004* | RRID:SCR_014222 | https://www2.mrc-lmb.cam.ac.uk/personal/pemsley/coot/ |
| Software, algorithm | Phenix.refine | *Adams et al., 2010* | RRID:SCR_014224 | https://www.phenix-online.org/documentation/reference/refinement.html |
| Software, algorithm | Graphpad prism 8.30 | Graphpad | RRID:SCR_002798 | https://www.graphpad.com/scientific-software/prism/ |
| Software, algorithm | Frealign | *Grigorieff, 2016* | RRID:SCR_016733 | https://grigoriefflab.umassmed.edu/frealign |
| Other | Ni-NTA Agarose | Qiagen | Cat. #: 30230 | Recombinant protein purification |
| Other | Lipofectamine 2000 | Thermo Fisher Scientific | Cat. #: 11668019 | Mammalia cell transfection |

