## [Decision Letter]

**Acceptance summary:**

This manuscript presents a cryo-EM structure of the tubulin tyrosine carboxypeptidase VASH1-SVBP bound to a microtubule, providing an explanation for its substrate specificity (microtubule vs soluble tubulin). Structure-guided mutagenesis combined with activity assays performed both in vitro and in cells confirm predictions made by the structure. This study provides new insight into the mechanism of substrate recognition by an important microtubule modifying enzyme.

**Decision letter after peer review:**

Thank you for submitting your article "Cryo-EM structure of VASH1-SVBP bound to microtubules" for consideration by *eLife*. Your article has been reviewed by three peer reviewers, one of whom is a member of our Board of Reviewing Editors, and the evaluation has been overseen by Vivek Malhotra as the Senior Editor. The reviewers have opted to remain anonymous.

The reviewers have discussed the reviews with one another and the Reviewing Editor has drafted this decision to help you prepare a revised submission.

Summary:

This manuscript presents a cryo-EM structure of VASH1-SVBP bound to 14-protofilament microtubule. The overall resolution is 3.1 A, but the resolution of VASH1-SVBP complex is lower. Nevertheless, the authors were able to dock in a previous crystal structure of VASH1-SVBP. Their structure revealed that VASH1-SVBP binds two adjacent α-tubulins in a microtubule, providing an explanation for its substrate specificity (microtubule vs soluble tubulin). The authors complement their structural observations with structure-guided mutagenesis combined with activity assays performed both in vitro and in cells. The study focuses on an important molecular player that is of interest to many in the microtubule field and addresses the important question of how the VASH1 enzyme recognizes its microtubule substrate. However, the quality of the study needs improvement, both technically as well as from a scholarship point of view. This will require additional evaluation/clarification of the EM data, a limited set of additional experiments and textual changes to be clearer about limitations, to cite references more appropriately, and to provide more detail for Materials and methods.

Essential revisions:

1) The majority of the figures showing interactions do not seem to match the structure that was included with the manuscript. The figures should all be redone with the final structure (or, conversely, the final structure should be submitted with the manuscript if the figures were generated later). As a consequence of this mismatch between the figures and the structure provided, some of the proposed contacts are too far away to be considered electrostatic interactions or hydrogen bonds. It is therefore not possible to evaluate the proposed interfaces without the appropriate structure.

a) List of interactions in question: K145 and E434 are 9A away in the structure. H268 and R390 are 6A apart. S269 and E386 are 4.4A apart, which is too far for a hydrogen bond as suggested. There's only 2 electrostatic interactions that are at 3A or less (R148 and E433; E271 and H309), while the rest are in a 4.5-5A range.

b) Additionally, the manuscript mentions that V270 is in contact with H309, E386, and E433, but in reality, only E386 is within contact range.

2) Given that the resolution of cryo-EM map is not uniform, the paper should include a local resolution plot/map, particularly given the lower resolution of the VASH1-SVBP density, which is the most important part of the structure (~6A vs 3.1A for the microtubule). Along the same lines, the current FSC calculation description was confusing and seemed to suggest that only the tubulin heterodimer was used for the calculation rather than also including the VASH1-SVBP. Was this the case? Clarity of this description should be improved.

3) "To build the structure of the microtubule-VASH1-SVBP complex, we docked the crystal structure of VASH1-SVBP and the cryo-EM structure of αβ-tubulin heterodimers of GMPCPP-stabilized microtubules into the EM density map as rigid bodies." PDB IDs for the X-ray structures and citations need to be provided.

4) Given that the major thrust of the manuscript is to provide a structural explanation for the substrate specificity of VASH1-SVBP, the proposed role of interface 3 should be strengthened by performing detyrosination assays with α/β tubulin dimers. Showing that the 3E mutant has no effect on detyrosination of tubulin dimers would be far more convincing than the lack of effect on the C-terminal tail peptide.

5) The section on the interaction between the C-term tail and VASH1-SVBP seems rather speculative given the low resolution of the map there and the data presented in the paper. This would be easier to judge if the authors included a locally-filtered map. Unless more convincing data/figures can be presented, it is recommend to move this section to the Discussion section. New figures with a clearer view of what is "unaccounted" density would be quite helpful in making this section less speculative. As the data is presented at the moment the reviewers conclude that the α tail is not really visible in the structure.

6) Figure 3C was considered unconvincing. Although the authors state in the text that "…. at interface 2, including R390 and H393, are not properly positioned for VASH1 binding", the differences in the figure seem small enough that minor adjustments of the side chain rotamers may restore those interactions. If the map is not good enough to see side chains in this region, the authors should limit themselves to showing a sphere at the position where the C α is. The conclusion drawn here should be either toned down/removed or more compelling evidence to support it should be provided, for example by testing whether mutating H393 and R390 on α-tubulin affects detyrosination of MTs but not of the dimer. That would provide more solid support for the idea proposed in Figure 3C.

7) Quantification: Functional assays (Figure 4) need to be quantified. Data shown in Figure 1C and Figure 4—figure supplement 1 should have statistical analysis.

8) The tubulin purification protocol that the authors employed (Ti et al., 2016) describes tubulin purification with an affinity tag on only β-tubulin. Using this procedure, results in a mixture of human and insect α-tubulin – can the authors clarify here how pure their tubulin is (the SDS gel will not resolve this). They might want to discuss whether tubulin heterogeneity might have contributed to the lower resolution of the enzyme and the difficulties to resolve the α-tubulin tail.

9) The microtubule reconstruction procedure should be described in more detail in the Materials and methods section.

10) Scholarship: The authors should please make sure that they cite the relevant literature correctly. Some guidance:

a) The authors state "The conformation of free αβ-tubulin heterodimers is different from that of αβ-tubulin in microtubules, with the latter being more compact and straight (Brouhard and Rice, 2018). There are also subtle differences between the conformations of α-tubulin in free αβ-tubulin heterodimers and microtubules." The authors should cite original studies on the topic (including work of one of the authors).

b) The authors state "We superimposed the N-terminal domain (residues 1-140) of α-tubulin in the free αβ-tubulin heterodimer onto the EM structure of the microtubule-VASH1-SVBP complex". It appears appropriate to cite Weinert et al., 2017 here, as the authors used their X-ray structure for comparison.

c) "…co-transfected theseVASH1 mutants with SVBP into HeLa cells (which had low endogenous levels of tubulin detyrosination)". A citation showing low levels of detyrosination in this cell line should be provided.

d) Figure 3C legend "Superimposition of the structure of α-tubulin from free αβ tubulin heterodimers (PDB:5nqu)" also lacks the citation.

e) The authors tend to cite only very recent literature. Key original papers should also be cited. Some of the citations are placed inappropriately. For example, in the Introduction paragraph two when citing the discovery of detyrosination, the appropriate reference is Hallak et al., 1977 only. The same applies for the discovery of TTL. When discussing MAPs that respond to the detyrosination/tyrosination, original studies should be cited and not only reviews (one of the reviews cited hardly even covers this). In paragraph four when discussing what TTL recognizes, does the Alushin et al. study really belong here? Also, the authors should cite the original Frealign paper in their Materials and methods.

---

## [Author Response]

Essential revisions:1) The majority of the figures showing interactions do not seem to match the structure that was included with the manuscript. The figures should all be redone with the final structure (or, conversely, the final structure should be submitted with the manuscript if the figures were generated later). As a consequence of this mismatch between the figures and the structure provided, some of the proposed contacts are too far away to be considered electrostatic interactions or hydrogen bonds. It is therefore not possible to evaluate the proposed interfaces without the appropriate structure.a) List of interactions in question: K145 and E434 are 9A away in the structure. H268 and R390 are 6A apart. S269 and E386 are 4.4A apart, which is too far for a hydrogen bond as suggested. There's only 2 electrostatic interactions that are at 3A or less (R148 and E433; E271 and H309), while the rest are in a 4.5-5A range.b) Additionally, the manuscript mentions that V270 is in contact with H309, E386, and E433, but in reality, only E386 is within contact range.

We thank the reviewer for catching this oversight. The figures were indeed made based on an earlier version of the structure, not the final version. We have now remade all figures based on the latest refined structures in the revised manuscript. More importantly, because of the lack of clear sidechain density for many key residues at the VASH1-microtubule interface, we have deleted all dashed lines that had indicated salt bridges and hydrogen bonds and rephrased the text to avoid making specific references to sidechains.

2) Given that the resolution of cryo-EM map is not uniform, the paper should include a local resolution plot/map, particularly given the lower resolution of the VASH1-SVBP density, which is the most important part of the structure (~6A vs 3.1A for the microtubule). Along the same lines, the current FSC calculation description was confusing and seemed to suggest that only the tubulin heterodimer was used for the calculation rather than also including the VASH1-SVBP. Was this the case? Clarity of this description should be improved.

We have added the local resolution map in Figure 2—figure supplement 1A in the revised manuscript. The original FSC calculation was indeed only for the tubulin heterodimer. Because the resolution of VASH1-SVBP is lower than that of tubulin heterodimer, we have calculated the FSC separately and have now included the FSC calculation of VASH1-SVBP in Figure 1—figure supplement 1C of the revised manuscript.

3) "To build the structure of the microtubule-VASH1-SVBP complex, we docked the crystal structure of VASH1-SVBP and the cryo-EM structure of αβ-tubulin heterodimers of GMPCPP-stabilized microtubules into the EM density map as rigid bodies." PDB IDs for the X-ray structures and citations need to be provided.

The PDB IDs for the structures of VASH1-SVBP and αβ-tubulin are 6OCG and 6DPU, respectively. This information, along with citations, has been provided in the revised manuscript.

4) Given that the major thrust of the manuscript is to provide a structural explanation for the substrate specificity of VASH1-SVBP, the proposed role of interface 3 should be strengthened by performing detyrosination assays with α/β tubulin dimers. Showing that the 3E mutant has no effect on detyrosination of tubulin dimers would be far more convincing than the lack of effect on the C-terminal tail peptide.

This is a great suggestion. We have performed the in vitro detyrosination assay on free αβ-tubulin heterodimers. The 3E mutation of VASH1 indeed has no effect on the detyrosination of αβ-tubulin heterodimers. These data are included in Figure 4C and Figure 4—figure supplement 1C of the revised manuscript.

5) The section on the interaction between the C-term tail and VASH1-SVBP seems rather speculative given the low resolution of the map there and the data presented in the paper. This would be easier to judge if the authors included a locally-filtered map. Unless more convincing data/figures can be presented, it is recommend to move this section to the Discussion section. New figures with a clearer view of what is "unaccounted" density would be quite helpful in making this section less speculative. As the data is presented at the moment the reviewers conclude that the α tail is not really visible in the structure.

We have postprocessed the EM density map with different programs by performing global or local B factor sharpening, with and without low-pass filtering. This extended C-terminal density is clear and continuous only when the map is low-pass filtered to 7 Å, indicating that this density is weak. In the revised version, we provide two different views of this density generated by Phenix.auto_sharpen, with local B factor sharpening and resolution cutoff at 7 Å in Figure 2D. Because this density is extremely weak, we have toned down our conclusions.

6) Figure 3C was considered unconvincing. Although the authors state in the text that "…. at interface 2, including R390 and H393, are not properly positioned for VASH1 binding", the differences in the figure seem small enough that minor adjustments of the side chain rotamers may restore those interactions. If the map is not good enough to see side chains in this region, the authors should limit themselves to showing a sphere at the position where the C α is. The conclusion drawn here should be either toned down/removed or more compelling evidence to support it should be provided, for example by testing whether mutating H393 and R390 on α-tubulin affects detyrosination of MTs but not of the dimer. That would provide more solid support for the idea proposed in Figure 3C.

We agree with the reviewer on this point. We have removed the original Figure 3C and the corresponding description in the revised manuscript.

7) Quantification: Functional assays (Figure 4) need to be quantified. Data shown in Figure 1C and Figure 4—figure supplement 1 should have statistical analysis.

We have added statistical analysis to Figure 1C and Figure 4—figure supplement 1 according to the reviewers’ suggestion. The p-values between WT and mutants were calculated using two-tailed student’s t-test.

8) The tubulin purification protocol that the authors employed (Ti et al., 2016) describes tubulin purification with an affinity tag on only β-tubulin. Using this procedure, results in a mixture of human and insect α-tubulin – can the authors clarify here how pure their tubulin is (the SDS gel will not resolve this). They might want to discuss whether tubulin heterogeneity might have contributed to the lower resolution of the enzyme and the difficulties to resolve the α-tubulin tail.

We (the Rice group) have previously shown that the recombinant human tubulin produced using the insect cell system contains human and insect α-tubulin molecules in roughly 70:30 proportion (see Figure 1D in PMID: 32077153). The Nogales and Kapoor labs have determined the structure of recombinant human microtubules bound to the motor domain of kinesin-1, and reported that there were no significant differences with the bovine brain microtubule structures at comparable resolutions. Thus, we do not believe that the lower resolution of VASH1 was caused by the heterogeneity of α-tubulin molecules. On the other hand, because the sequence of the C-terminal tail of the insect α-tubulin is different from that of human α-tubulin, it is possible that this heterogeneity contributes to the difficulty to resolve the α-tubulin tail. We have discussed this possibility in the revised manuscript.

9) The microtubule reconstruction procedure should be described in more detail in the Materials and methods section.

We have now described the microtubule reconstruction procedure in more detail in the revised manuscript.

10) Scholarship: The authors should please make sure that they cite the relevant literature correctly. Some guidance:a) The authors state "The conformation of free αβ-tubulin heterodimers is different from that of αβ-tubulin in microtubules, with the latter being more compact and straight (Brouhard and Rice, 2018). There are also subtle differences between the conformations of α-tubulin in free αβ-tubulin heterodimers and microtubules." The authors should cite original studies on the topic (including work of one of the authors).b) The authors state "We superimposed the N-terminal domain (residues 1-140) of α-tubulin in the free αβ-tubulin heterodimer onto the EM structure of the microtubule-VASH1-SVBP complex". It appears appropriate to cite Weinert et al., 2017 here, as the authors used their X-ray structure for comparison.

We apologize for not citing the original research articles in this section. Because this section and Figure 3C have been deemed to be too speculative (see comment 6 above), we have decided to remove this section from the revised manuscript. Therefore, it is no longer necessary to cite the references alluded to by the reviewers. We will, however, pay more attention to proper literature citing in future studies.

c) "…co-transfected theseVASH1 mutants with SVBP into HeLa cells (which had low endogenous levels of tubulin detyrosination)". A citation showing low levels of detyrosination in this cell line should be provided.

We have cited the original reference.

d) Figure 3C legend "Superimposition of the structure of α-tubulin from free αβ tubulin heterodimers (PDB:5nqu)" also lacks the citation.

Again, we have removed this figure panel in the revised manuscript. The citation is no longer necessary.

e) The authors tend to cite only very recent literature. Key original papers should also be cited. Some of the citations are placed inappropriately. For example, in the Introduction paragraph two when citing the discovery of detyrosination, the appropriate reference is Hallak et al., 1977 only. The same applies for the discovery of TTL. When discussing MAPs that respond to the detyrosination/tyrosination, original studies should be cited and not only reviews (one of the reviews cited hardly even covers this). In paragraph four when discussing what TTL recognizes, does the Alushin et al. study really belong here? Also, the authors should cite the original Frealign paper in their Materials and methods.

As suggested, we have only cited the original studies when referring to the discoveries of detyrosination and TTL. We have cited original studies when discussing MAPs that respond to detyrosination/tyrosination. We have removed the Alushin et al. citation when discussing TTL recognition of tubulin. We have also cited the original Frealign paper in the Materials and methods.